# Characterization of Yeasts Isolated from Parmigiano Reggiano Cheese Natural Whey Starter: From Spoilage Agents to Potential Cell Factories for Whey Valorization

**DOI:** 10.3390/microorganisms9112288

**Published:** 2021-11-03

**Authors:** Serena Martini, Mattia Bonazzi, Ilaria Malorgio, Valentina Pizzamiglio, Davide Tagliazucchi, Lisa Solieri

**Affiliations:** 1Department of Life Sciences, University of Modena and Reggio Emilia, Via Amendola, 2-Pad. Besta, 42122 Reggio Emilia, Italy; serena.martini@unimore.it (S.M.); 240824@studenti.unimore.it (M.B.); malorgioilaria.94@gmail.com (I.M.); davide.tagliazucchi@unimore.it (D.T.); 2Consorzio del Formaggio Parmigiano Reggiano, via J.F. Kennedy 18, 42124 Reggio Emilia, Italy; pizzamiglio@parmigianoreggiano.it

**Keywords:** Parmigiano Reggiano cheese, whey, natural whey starter, ethanol, bioactive peptides, yeasts, *Kluyveromyces marxianus*, *Wickerhamiella pararugosa*, *Torulaspora delbrueckii*

## Abstract

Whey is the main byproduct of the dairy industry and contains sugars (lactose) and proteins (especially serum proteins and, at lesser extent, residual caseins), which can be valorized by the fermentative action of yeasts. In the present study, we characterized the spoilage yeast population inhabiting natural whey starter (NWS), the undefined starter culture of thermophilic lactic acid bacteria used in Parmigiano Reggiano (PR) cheesemaking, and evaluated thermotolerance, mating type, and the aptitude to produce ethanol and bioactive peptides from whey lactose and proteins, respectively, in a selected pool of strains. PCR-RFLP assay of ribosomal ITS regions and phylogenetic analysis of 26S rDNA D1/D2 domains showed that PR NWS yeast population consists of the well-documented *Kluyveromyces marxianus*, as well as of other species (*Saccharomyces cerevisiae*, *Wickerhamiella pararugosa*, and *Torulaspora delbrueckii*), with multiple biotypes scored within each species as demonstrated by (GTG)_5_-based MSP-PCR. Haploid and diploid *K. marxianus* strains were identified through *MAT* genotyping, while thermotolerance assay allowed the selection of strains suitable to grow up to 48 °C. In whey fermentation trials, one thermotolerant strain was suitable to release ethanol with a fermentation efficiency of 86.5%, while another candidate was able to produce the highest amounts of both ethanol and bioactive peptides with potentially anti-hypertensive function. The present work demonstrated that PR NWS is a reservoir of ethanol and bioactive peptides producer yeasts, which can be exploited to valorize whey, in agreement with the principles of circularity and sustainability.

## 1. Introduction

In a hungry world where the human population continues to rise and biological resources are limited, food byproduct valorization can contribute to provide sustainable solutions to the increasing food demand [1]. According to Ellen MacArthur Foundation’s butterfly diagram [2], food waste and byproduct valorization can also positively impact the environment by reducing greenhouse gas emissions [3].

The dairy industry is a leader in the European food industry, and it is the second-largest sector of food waste generation. Whey is the liquid residue resulting from the removal of the high molecular weight milk proteins, such as caseins, during the production of cheese and other dairy products [4]. Whey is the most important byproduct of cheesemaking as it represents 85–90% of the original milk volume [5]. It is estimated that 9 million tons of cheese is produced within the EU per annum. This generates around 50 million m^3^ of whey [6]. The high content of lactose (44–46 g/L) and residual proteins, especially low molecular proteins (2.5–10 g/L based on the kind of whey), increase chemical (from 60–80 g/L) and biological (30–50 g/L) oxygen demand values (BOD and COD, respectively) other than the limits established by national and international standards and make whey disposal in the environment very expensive. On the other hand, lactose and proteins represent potential reusable substances Traditionally whey has been used for animal feeding and ricotta cheese production. Recently, in agreement with sustainability and circularity principles, novel attempts were made to convert whey into value-added bioproducts. Whey is used to produce protein powders by spray-drying for infant nutrition, sport formulas and food additives, leaving the lactose rich liquid, called whey permeate, for environmental disposal. Several other bio-products can be produced from whey and whey permeate, such as bioethanol [7], biohydrogen [8,9], prebiotics [10], bacteriocin [11,12], biopolymer [13], and exopolysaccharides [14]. Additionally, whey permeate has been investigated as cheap medium to produce single cell proteins [15,16] and to propagate either probiotics or other industrially significant cell factories [17]. More recent efforts demonstrated that whey can be a source of bioactive compounds such as bioactive peptides, which can be released from whey proteins by the action of either enzymes or microbes [5,18,19,20,21]. Bioactive peptides are defined as specific protein fragments that positively impact body functions and can promote good health and the prevention of diseases [22]. Whey drink or whey protein hydrolysates enriched in bioactive peptides can valorize the whey, creating value-added products that may promote human health. Integration of these solutions in a rationale frameshift for valorizing different whey components is called the whey biorefinery concept [23].

Many of the attempts for whey valorization entail the use of microbes as cell factories for catalyzing whey bioprocessing. Among them, non-conventional yeasts alternative to the well-known model organism *Saccharomyces cerevisiae*, are attracting increasing interest to create value-added products from food waste thanks to their ability to ferment alternative sugars other than glucose and to survive under multiple stressors better than *S. cerevisiae* [24]. Consequently, different naturally fermented foods and environmental niches have been exploited as reservoirs of wild non-conventional yeasts.

Parmigiano Reggiano (PR) is an Italian protected designation of origin (PDO) hard and cooked cheese made from raw partly skimmed cow’s milk supplemented with natural whey starter [25,26]. Natural whey starter (NWS) is an undefined consortium of thermophilic lactic acid bacteria (LAB) obtained by incubation of the previous day residual acidic whey at a gradually decreasing temperature after the curd heating [27,28,29]. Obligate homofermentative LAB belonging to *Lactobacillus helveticus*, *Streptococcus thermophilus*, and *Lactobacillus delbrueckii* (especially subspecies *lactis*) and the heterofermentative species *Limosilactobacillus fermentum* dominate the NWS microbiota and are responsible for fast milk acidification and casein precipitation together with the calf rennet. Additionally, lactose-fermenting yeasts can contaminate NSW and co-exist with starter LAB [30]. In PR NWS *Kluyveromyces marxianus* was the dominant yeast species [30,31]. This food-grade yeast is an emerging cell factory proposed as bioethanol producer [32], probiotic agent [33], and enzyme-producer [34].

In this study, we explored yeast biodiversity of PR NWS and evaluated yeast candidates to produce ethanol and to release bioactive peptides from whey proteins during whey fermentation. We demonstrated that yeasts from PR NWS are not only spoilage agents but also beneficial cell factories exploitable for ethanol production and development of whey protein hydrolysates and whey fermented drink enriched in bioactive peptides.

## 2. Materials and Methods

### 2.1. Chemicals and Reference Strains 

All chemicals and media were purchased from Sigma Aldrich (St. Louis, MO, USA), except where differently indicated. Oligonucleotides were from Bio-Fab Research (Rome, Italy). *Kluyveromyces marxianus* CBS608 was used as reference *MAT*a/*MAT*α strain and cultivated in YPDA (1% *w*/*v* yeast extract, 2% *w*/*v* peptone, 2% *w*/*v* dextrose, and 2% *w*/*v* agar) medium.

### 2.2. Sampling, Physicochemical and Microbiological Analyses 

NWS samples were obtained from three different PR cheese dairies located in the PDO cheese production area (Reggio Emilia, Italy). Fifty milliliters of NWS were collected in duplicates in sterile flasks just before the addition to the vat milk (a mixture of morning milk and partially skimmed evening milk). Samples were shipped to the laboratory under refrigerated conditions and immediately submitted to physicochemical and microbiological determinations. Titratable acidity was determined using the Soxhlet–Henkel method with 0.25 N NaOH [35]. The pH was determined at samples temperature of 25 °C after calibration of pH meter (Crison Instruments, Barcelona, Spain) at the same temperature. Lactic acid bacteria counts were determined by fluorocytometry with a Bactoscan 8000 apparatus (Foss Electric, Hillerød, Denmark) and expressed as Log_10_ CFU/mL values.

Yeast counts were determined by plating ten-fold diluted NWS samples on YPDA and YPLA (1% *w*/*v* yeast extract, 2% *w*/*v* peptone, 2% *w*/*v* lactose, and 2% *w*/*v* agar) media through spreading method. To inhibit bacterial growth, both media were supplemented with 100 mg/L chloramphenicol. The YPDA and YPLA plates were incubated at 28 °C and 42 °C for 48–72 h, respectively, as previously reported [31]. Analyses were carried out in triplicate. Yeast population was expressed as mean of Log_10_ CFU/mL values. At least 10% of colonies recovered from 20–200 CFU/mL plates were randomly selected from each medium and purified by streaking at least two times onto the same medium. Purified isolates were maintained at 4 °C on YPDA slants and stored at −80°C in the YPD broth medium supplemented with 25% (*w*/*v*) glycerol as cryopreservation agent.

### 2.3. DNA Extraction, Molecular Caracterization, and Phylogenetic Analysis

For DNA isolation, pure cultures of yeasts were grown in YPD broth at 28 °C for 48 h. Genomic DNA (gDNA) was extracted as reported by Hoffman and Winston [36]. The final quantity of the resultant DNA was determined by NanoDrop ND-1000 device (Thermo Scientific Waltman, MA, USA) and diluted to 50 ng/μL with sterile ultra-pure water. Template DNA samples were stored at −20 °C till used. 

PCR amplification of ITS regions was carried out with the primers ITS1 (5′-TCCGTAGGTGAACCTGCGG-3′) and ITS4 (5′-TCCTCCGCTTATTGATATGC-3′) [37] in a reaction mixture (final volume 40 μL) containing 1X Dream Taq Green Buffer (Thermo Scientific Waltman, MA, USA), 1.5 mM MgCl_2_, 0.2 mM of each dNTPs (Thermo Scientific Waltman, MA, USA), 0.5 µM of each primer, 0.5 U of DreamTaq DNA polymerase (Thermo Scientific Waltman, MA, USA) and 50 ng of gDNA. Thermal conditions consisted of 94 °C initial denaturation for 1 min; 35 amplification cycles of 1 min at 94 °C, 2 min at 55 °C, 2 min at 72 °C; final extension at 72 °C for 10 min. 

PCR amplification of the D1/D2 domains of the 26S rRNA gene (LSU) was carried out with the primers NL-1 (5′-GCATATCAATAAGCGGAGGAAAAG-3′) and NL4 (5′-GGTCCGTGTTTCAAGACGG-3′) [38] using rTAq DNA polymerase under the same reaction conditions reported for ITS region. Thermal cycling included initial denaturation at 94 °C for 5 min; 36 cycles of 94 °C for 1 min, 52 °C for 45 s, 72 °C for 2 min; final extension at 72 °C for 10 min, followed by cooling at 4 °C.

All PCR reactions were carried out in a T100 thermal cycler (Bio-Rad, Hercules, CA, USA). The presence of amplicons was confirmed by electrophoresis in 1.2% (*w*/*v*) agarose gel in 0.5× TBE (89 mM Tris-borate, 2 mM EDTA, pH 8) buffer and stained with 0.5 µg/mL of ethidium bromide. All gels were visualized by UV and captured as TIFF format files by in a gel documentation system (BioDocAnalyze, Biometra, Göttingen, Germany).

ITS amplicons were subjected to restriction fragment length polymorphism (RFLP) analysis using endonucleases *Hae*III and *Hinf*I (Thermo Scientific, Waltman, MA, USA), according to manufacturer’s instructions. The resulting restriction fragments were separated by 2.0% (*w*/*v*) agarose gel electrophoresis in 0.5× TBE buffer at 70 V for 2 h. PCR amplicons and restriction profiles were compared to those present in the Yeast-ID database (www.yeast-id.org, accessed on 9 September 2021) with rank parameter at ±20 bp.

PCR amplicons of 26S rDNA D1/D2 domains were purified using DNA Clean & Concentrator™-5 Kit (Zymo Research, Orange, CA, USA) and sequenced on both strands using NL1 and NL4 primers through a DNA Sanger dideoxy sequencing process performed by Bio-Fab Research (Rome, Italy). Consensus sequences were merged using the program SeqMan (DNASTAR, Madison, WI, USA) and the poor-quality ends were edited manually to remove primers. The nucleotide sequences were compared with sequences available in the NCBI (www.ncbi.nih.gov; last access on 3 September 2021). Strains with 0–3 nucleotide differences in the D1/D2 domain were treated as conspecific, while strains showing greater than 1% nucleotide substitutions were considered to belong to different species [38]. The related sequences were aligned with Muscle program [39] in MEGA X software [40] and the resulting alignment was subjected to a DNA substitution model analysis to select the best-fitting model. Phylogenetic relationships were inferred using the Kimura 2-parameter (K2P) model and the neighbor joining (NJ) method. Among sites rate variation was modelled by a gamma distribution (+G). Bootstrap support values were obtained from 1000 random resamplings. Tree was visualized using Interactive Tree of Life (ITOL) [41] and rooted at outgroup reference strains. The sequences obtained in this study were deposited in the GenBank NCBI database with the accession numbers MZ491084-MZ491094.

### 2.4. Genotyping

Intraspecific diversity of isolates was assessed by microsatellite-primed PCR (MSP-PCR) using the primer (GTG)_5_ (5′-GTGGTGGTGGTGGTG-3′), as previously reported [42]. PCR products were separated by 1.8% (*w*/*v*) gel electrophoresis in 0.5× TBE buffer for 7 h under constant 70 V under refrigerated conditions. The GeneRuler 100 bp Plus DNA Ladder (ThermoScientific, Waltman, MA, USA) served as a molecular size marker. All DNA fingerprint digital images were captured in Tiff format as reported above and processed by the BioNumerics software version 3.0 (Applied Maths, Sint-Martens-Latem, Belgium). Band assignment was manually curated after automatic band detection. Bands patterns similarities were calculated using the Pearson’s correlation similarity coefficient. Optimization value and curve smoothening were set at 0.5%. Dendrograms were constructed by the unweighted-pair group method using the arithmetic means (UPGMA) clustering method. Isolates of ≥92% similarity were treated as a single strain.

### 2.5. MAT Genotyping of Kluyveromyces marxianus Strains

*MAT* genotyping was carried out with the primers SLA2 (5′-TATACATGGGATCATAAATC-3′) [43], MATa1D (5′-GGTTTGGCAGGAGTACAACTA-3′), and MATa1D (5′-TGAAATCCAAAGCACCAACT-3′) [44]. PCR reactions were performed in a final volume of 20 μL containing 1X Green Buffer (Thermo Scientific, Waltman, MA, USA); 200 μM dNTPs (Thermo Scientific, Waltman, MA, USA), 0.5 μM each primer, 0.5 U Dream Taq DNA polymerase (Thermo Scientific, Waltman, MA, USA) and 50 ng of gDNA. PCR amplification was performed on a MyCycler TM thermal cycler (BioRad, Hercules, CA) with an initial denaturation at 95 °C for 3 min, followed by 10 cycles consisting of 30 s at 95°C, 45 s at 53 °C and 3 min at 72 °C, and 25 cycles consisting of 30 s at 95 °C, 45 s at 55 °C and 3 min at 72 °C, and a final extension of 10 min at 72 °C. A single primer test was done to exclude any unspecific amplifications. PCR products were separated by 1.5% (*w*/*v*) agarose gel electrophoresis in 0.5 X TBE buffer for 4 h at 70 V. DNA fragment size was estimated using Gene Ruler^TM^ 1 kb Plus DNA ladder (Thermo Scientific, Waltham, MA). Expected sizes were 2620 and 2863 bp for *MAT*a and *MAT*α amplicons, respectively [44].

### 2.6. Thermotolerance Assay

The ability to grow at 42, 45 and 48 °C was tested as previously reported [45]. Briefly, four ten-fold dilutions from 10^2^ to 10^5^ cell/mL were prepared from overnight grown cultures of each yeast cultures (27 °C in 5 mL of YPD broth). Five microliters of cell suspensions were spotted onto YPDA agar plates pre-adapted to the appropriate temperature. Plates were sealed with Parafilm and incubated for 5 days at the appropriate temperature. Plates incubated at 27 °C were used as positive control. Assays were performed in duplicates.

### 2.7. Whey Fermentation Assays

The assays were done according to Tofalo et al. [46] with a few modifications. Briefly, yeast strains grown overnight at 28 °C in 30 mL YPD medium were inoculated into 100 mL bottles filled with 50 mL of pasteurized (95 °C for 10 min) cow acidic whey (5.67% *w*/*v* ± 0.06 lactose, 0.05% *w*/*v* ± 0.02 glucose, 0.65% *w*/*v* ± 0.03 galactose and 9.5% *w*/*v* ± 0.1 lactic acid; pH 4.80 ± 0.07) at the final concentration of 5 x 10^7^ CFU/mL. Inoculated whey was overlaid with 10 mL of paraffin oil and fermentation was carried out at 30 °C. Non inoculated whey was used as negative control. Weight loss of the flasks due to CO_2_ release was measured with an analytical balance and used to monitor fermentation progress. Fermentation trials were stopped after 14 days, and supernatants were collected by centrifugation at 14,000 rpm for 20 min at 4 °C and then stored at −20 °C until further analysis. Lactose consumption and ethanol production were enzymatically assessed (Megazyme Bray, Ireland).

### 2.8. Proteolysis Degree and Peptidomic Analysis by Ultra High Performance Liquid Chromatography/High Resolution Mass Spectrometry (UHPLC/HR-MS)

Proteolysis degree was determined in the supernatants obtained from the whey fermentation trials, by the 2,4,6-trinitrobenzenesulphonic acid (TNBS) method [47]. The results were expressed as mmol/L of leucine equivalents. All the analyses were carried out in duplicates.

Low molecular weight peptides released in the supernatants collected from the whey fermentation experiments were separated through a C18 column (Acquity UPLC HSS C18 Reversed phase, 2.1 × 100 mm, 1.8 μm particle size, Waters, Milan, Italy) by using a UHPLC system (UHPLC Ultimate 3000 separation module, Thermo Scientific, San Jose, CA, USA) and analyzed by a Q Exactive Hybrid Quadrupole-Orbitrap Mass Spectrometer (Thermo Scientific, San Jose, CA, USA). The full description of the chromatographic conditions, mass spectrometry and tandem mass spectrometry parameters was reported in Martini et al. [48].

Peptide sequencing was carried out by using MASCOT (Matrix Science, Boston, MA, USA) protein identification software as fully described in Martini et al. [48]. The full lists of identified peptides were submitted to the Milk Bioactive Peptides Database (MBPDB) to identify peptides with 100% homology to previously demonstrated bioactive peptides [49].

### 2.9. Statistical Analysis

All data were presented as mean ± standard deviation (SD) for three replicates for each sample. The growth kinetics of yeasts in whey were calculated with Grofit package implemented in R [50]. Two-way ANOVA with Bonferroni post-test were performed using Graph Pad Prism (GraphPad Software, San Diego, CA, USA). The differences were considered significant with *p* value < 0.05.

## 3. Results

### 3.1. Physicochemical Analysis and Microbiological Counts

Physicochemical parameters and microbiological counts of NWS samples were reported in Table 1. Yeast population ranged from 2.03 to 3.43 Log_10_ CFU/mL in R_NWS and L_NWS samples, respectively. Except for R_NWS, no significant differences in yeast counts were detected between culture conditions. Sample L_NWS showed higher yeast population than R_NWS and C_NWS (*p* < 0.05). Titratable acidity measured as SH degree of L_NWS was intermediated between R_NWS and C_NWS, while pH value in L_NWS was higher than those found in samples R_NWS and C_NWS (*p* < 0.05).

### 3.2. Yeast Molecular Characterization and Species Assignment

Ninety-one isolates retrieved from YPDA and YPLA media at 28 and 42 °C, respectively, were submitted to PCR-RFLP analysis of ITS regions with the endonucleases *Hae*III and *Hinf*I. We identified 4 restriction patterns, referred to as A to D (Table 2). Search in Yeast-ID database was carried out for a first tentative species attribution. Pattern A matched *Kluyveromyces dobzhanskii* (100% matching), followed by *K. marxianus* (94% matching), while patterns B and D matched *S. cerevisiae* and *Torulaspora delbrueckii* profiles (100% matching), respectively. Pattern C was shared by five *Candida* spp., including *Candida incommunis*, *Candida pararugosa* (current name *Wickerhamiella pararugosa*), *Candida pseudointermedia*, *Candida intermedia*, *Candida catenulata* (current name *Diutina catenulata*), and *Candida diversa* (current name *Saturnispora diversa*).

To assist the species identification by ITS PCR-RFLP, representative strains for each restriction pattern were submitted to sequencing of D1/D2 domain of 26S rRNA gene. A total of 11 sequences were submitted to BLASTn search against Refseq database to identify the closest relatives of the sequenced strains. A dataset of 33 sequences were built, aligned and phylogenetic relationships were inferred by NJ method. As shown in Figure 1, the sequences were resolved into four distinct clades. Strains RO201 and CA102 placed into *Saccharomyces* clade and formed a monophyletic group with *S. cerevisiae* NRRL Y-12632, in agreement with ITS PCR-RFLP analysis. Therefore, they were identified as *S. cerevisiae*. The sequences of strains representatives of pattern A aligned closely with those of *K. marxianus* strains CBS 712 and NRRL Y-8281, while strain RO204_3 grouped together with the type strain of *T. delbrueckii* with robust support. These results agreed with those found by ITS PCR-RFLP analysis. Finally, phylogenetic analysis identified strains with restriction pattern C as *W. pararugosa*. However, strains LA118 and LA206 showed 6 SNPs compared to *W. pararugosa* CBS 1010 (98.94% identity) (Appendix A). Analyses with additional molecular barcodes are required to further investigate taxonomic position of these strains.

### 3.3. Yeast Diversity and Species Distribution

PCR-fingerprinting with primer (GTG)_5_ was carried out to assess genetic diversity of 82NWS isolates, including 3 *T. delbrueckii*, 3 *S. cerevisiae*, 55 *K. marxianus*, and 21 *W. pararugosa* strains. Complex fingerprinting patterns consisted of 5 to 11 fragments with size from 442 to 2884 bp (Figure 2). The UPGMA dendrogram created using Pearson similarity coefficient grouped strains into two major clusters with a similarity level of 26.3%. Major cluster I grouped *S. cerevisiae* and *T. delbrueckii* strains, while major cluster II the *K. marxianus* and *W. pararugosa* strains. *S. cerevisiae* and *T. delbrueckii* strains branched at similarity level of 32.4% within cluster I, while most of *W. pararugosa* strains clustered separately from *K. marxianus* strains. Application of a similarity threshold of 92% as reproducibility cut-off allowed the identification of 12 different genotypes and 10 singletons (Figure 2). Most of the isolates clustered congruently with the dairy provenience, species attribution and isolation conditions. Except for LA218 and RO203, 19 *W. pararugosa* isolates clustered together and divided them into two different genotypes (similarity threshold of 92%). Most of *K. marxianus* strains clustered together with a similarity level of 76.8%. We scored five different genotypes and five singletons in this clusters. The only exceptions were *K. marxianus* LA102, LA110, CA111, and CA208, which branched separately into two distinct clusters. Based on the genotypes scored, a pool of 24 *K. marxianus* strains was selected for subsequent functional characterization.

Considering the strains isolated in this study, *K. marxianus* was the species with the highest occurrence (61.5%), followed by *W. pararugosa* (23.1%), *S. cerevisiae* (12.1%) and *T. delbrueckii* (3.3%) (Figure 3, panel A). Remarkably, *W. pararugosa* and *T. delbrueckii* have been never described in PR NWS. *K. marxianus* and *S. cerevisiae* were ubiquitous in all the samples considered, while *T. delbrueckii* was detected only in R_NWS (Figure 3, panel B). As expected, *K. marxianus* was dominant in R_NWS and C_NWS, whereas L_NWS exhibited *W. pararugosa* as dominant species. In this sample four different genotypes of *W. pararugosa* were detected. Intrastrain diversity was high also in *K. marxianus* populations with 3, 7, and 3 different genotypes detected in R_NWS, C_NWS, and L_NWS samples, respectively (Figure 3, panel B).

### 3.4. MAT Genotyping and Thermotollerance Assays

The mating type defines cell identity and competence for mating (in either *MAT*a or *MAT*α cells) and meiosis (in *MAT*a/*MAT*α cells) [51]. Furthermore, *MAT* genotyping could be informative on ploidy of yeast cells, with either *MAT*a or *MAT*α locus suggesting a haploid status, while *MAT*a/*MAT*α supporting a diploid status. Therefore, we firstly characterized 24 *K. marxianus* strains for their *MAT* genotype. Multiplex PCR assay targeting region across *SLA2*-*MAT* junctions (outside the Z regions) allowed for the discrimination of 50% of strains as *MAT*a/*MAT*α, 33.33% as *MAT*α, and 16.67% as *MAT*a (Figure 4). The recovery of putative haploid *MAT*a and *MAT*α isolates could be useful for future breeding programs.

Among stressful conditions, high temperature is frequently encountered in industrial bioprocesses. Interestingly, the tolerance to high temperature is a remarkable feature in *K. marxianus*, which can ferment sugars into ethanol at temperature above 40 °C, with some strains suitable to growth until 52 °C [52]. Therefore, we screened 24 *K. marxianus* candidates for the ability to grow at 45 and 48 °C. We firstly identified 13 candidates suitable to grow at 45°C. Among them, five strains, namely CA204, CA205, CA116, LA202, and LA112, were also able to grow up to 48 °C (Figure 5).

### 3.5. Whey Fermentation

Twelve *K. marxianus* strains representative of different temperature scores and with different mating type (namely 7 *MAT*a/*MAT*α, 4 *MAT*a, and 1 *MAT*a) were randomly chosen to ferment whey. Two *W. pararugosa* (LA118 and LA218) and one *T. delbrueckii* (RO204-3) strains were also considered for comparative purposes. Growth in whey was monitored as weight loss over time and the resulting CO_2_ evolution trends were modeled in Grofit [50]. Fermentation trials in whey resulted in five curves fitted with Gompertz equation (CA111, LA102, LA210, LA118, and LA218), one curve with modified Gompertz equation (strain RO204-3), six curves with Richards model (strains CA104, CA105, CA116, CA204, CA213, and RO101) and the remaining three strains with logistics (strains CA207, CA214, LA202). Three characteristic parameters, namely λ (lag phase), μ (growth rate), A (representing the maximum cell growth), were calculated from the model that best fit the data (Appendix A). As reported in Figure 6A *K. marxianus* CA213 and LA102 showed the highest growth rate in whey followed by strains CA207 CA111 and CA214, while *W. pararugosa* LA218 the lowest one (*p* < 0.05). *T. delbrueckii* RO204-3 started to release CO_2_ very late compared to other strains, resulting in low growth rate (*p* < 0.05).

The ethanol production and lactose consumption were determined at the end of whey fermentation trials. All the tested *K. marxianus* strains were able to convert lactose into ethanol, with strain CA214 being the most productive one (32.45 ± 9.18 g/L of ethanol produced) followed by CA104, CA116, CA213, and CA207, which were able to release about 26–28 g/L of ethanol in the medium. During whey fermentation by *K. marxianus*, the increase in ethanol production was accompanied by a disappearance in lactose concentration, which was undetectable or near to zero at the end of the fermentation. During alcoholic fermentation, the maximum ethanol production from hexoses (glucose and galactose) and lactose is 0.51 and 0.53 g/g, respectively. The total amount of lactose in whey was 56.7 g/L whereas the total amount of hexoses (glucose and galactose) was 7.0 g/L. Taking into account these data, the maximum ethanol production should be 30.1 g/L from lactose and 3.6 from hexoses (33.7 g/L of ethanol maximum production). In the case of *K. marxianus* strain CA214, the final ethanol concentration was 32.45 g/L, suggesting an almost complete conversion of sugars in ethanol (96.3% of fermentation efficiency). In the case of *K. marxianus* CA104, CA116, CA213, and CA207 the calculated ethanol efficiency ranged from about 77 to 83%. Previous studies reported ethanol efficiency of more than 70% for *K. marxianus* strains [53].

Lactose consumption was correlated with CO_2_ production and ethanol release for most of the *K. marxianus* strains, but not for *W. pararugosa* LA218 and *T. delbrueckii* RO204-3 (Figure 6B,C). In these strains, the slight consumption of lactose suggests the presence of β-galactosidase activity. Accordingly, Borelli et al. [54] and Andrade et al. [55] also reported *T. delbrueckii* dairy strains with β-galactosidase activity as specialized trait related to adaptation to isolation niche. However, in our strains, the ethanol production greatly exceeded the theoretical amount expected both from hexoses and lactose consumption. We can speculate a possible alternative pathway for ethanol production in these strains, such as the conversion of alanine or other amino acids into pyruvate, but further analyses are required to elucidate this point. In addition, for both these strains the low release of CO_2_ during the fermentation trials resulted very low growth rate (Figure 6A). We can speculate that the alleged lack of CO_2_ production might be an artifact from the gravimetric measurements, which can be masked by the strong increase in the biomass during fermentation.

### 3.6. Proteolysis and Bioactive Peptide Characterization

Dairy yeasts exhibit proteolytic behavior [56] and thus the potential for releasing bioactive peptides. This phenotype has been largely investigated in milk but poorly in whey. Recently, native microbiota of whey was proven to release bioactive peptides, but yeasts and LAB responsible for this proteolysis have been not characterized. In other works, *K. marxianus* and *Debaryomyces hansenii* generated antihypertensive peptides from the whey α-lactalbumin and β-lactoglobulin, alone [57,58] or in combination with LAB [59], but all these data arose from purified proteins and not from raw whey. Therefore, we determined whether whey proteins have been hydrolyzed by yeast fermentation. Figure 7 shows that the most proteolytic *K. marxianus* strains were CA105 and CA116 (amount of released free amino group of 17.0 ± 1.0 and 16.1 ± 0.3 mmol/L of leucine equivalent, respectively). The only *K. marxianus* strain without any proteolytic activity was CA207. *W. pararugosa* and *T. delbrueckii* strains also showed proteolytic phenotype, with *T. delbrueckii* RO204-3 showing the highest activity (Figure 7).

The most proteolytic strains were selected for further high-resolution mass spectrometry analysis with the aim to identify the peptide profile. In particular, the samples analyzed were from whey fermented by *T. delbrueckii* RO204-3 and *K. marxianus* CA105, CA116, CA214, LA202, and RO101. The full list of identified peptides and the full MS data are reported in Appendix A. Strain RO101 was the highest peptide producer (141 peptides), followed by CA214 (119 peptides) and CA105 (109 peptides). *T. delbrueckii* RO204-3 released the lowest number of peptides (75) (Figure 8). Yeasts proteases were active towards whey proteins especially β-lactoglobulin and, to a lesser extent, α-lactalbumin. However, *K. marxianus* strains mainly hydrolyzed β-casein (Figure 8). Cheese whey used for fermentation trials derived from PR cheese production after casein clogging and precipitation. It contained a prevalence of serum proteins but also a minor part of non-precipitated caseins or larger water-soluble casein peptides, released by the action of endogenous milk proteases, chymosin or starter LAB, which remained in the whey during cheese manufacturing [60,61]. Despite their presence in higher amounts in whey, serum proteins were less hydrolyzed than β-casein. Differently from caseins, which are characterized by a flexible and open structure, whey proteins displayed a rigid globular structure recalcitrant to the hydrolysis by proteases [62,63].

Strains CA105 and RO101 were the most active towards whey proteins β-lactoglobulin and α-lactalbumin (40 and 37 whey derived peptides, respectively). Comparison of peptide patterns revealed a low number of shared peptides (14), a signal of a great diversity of proteases responsible for hydrolytic process (Figure 9). As expected from different taxonomic position, *T. delbrueckii* RO204-3 showed 33 exclusive peptides (corresponding to the 44% of peptides identified), whereas five *K. marxianus* strains shared 35 peptides. The strains most similar in peptide profiles were CA214 and RO101 with 87 common peptides. Differently from the others, these strains were able to extensively hydrolyze the N-terminal region of β-casein and the C-terminal region of κ-casein suggesting a similar specificity of the proteases involved in the hydrolytic process.

Search in Milk Bioactive Peptides DataBase (MBPDB) allowed the identification of 22 bioactive peptides (100% homology) (Table 3). Most of the bioactive peptides came from β-casein (13 bioactive peptides) and αS1-casein (four bioactive peptides) hydrolysis. Most of the identified bioactive peptides were angiotensin I-converting enzyme (ACE) inhibitors (13 peptides), anti-microbial (five peptides), and antioxidant (five peptides). The strain that produced the highest number of bioactive peptides was CA214 (14 bioactive peptides) followed by RO101 (10 bioactive peptides). Strain CA214 also released the highest number of ACE-inhibitory peptides (10 peptides) followed by RO101 (eight peptides) and CA105 (eight peptides).

## 4. Conclusions

Yeasts inhabiting NWS have been conventionally considered detrimental microorganisms which divert sugars from conversion to lactic acid by the action of starter LAB. Consequently, this microbial fraction has been considered neglectable so far. Here we demonstrated that PR NWS is a valuable reservoir of potential yeast cell factories to valorize whey through the fermentative production of ethanol and bioactive peptides. Yeast population in PR NWS is more complex than previously reported with more identified species other than the previously documented *K. marxianus*, such as *T. delbrueckii*, *W. pararugosa*, and *S. cerevisiae*. We also found that more than one biotype is present within each dominant species. Interestingly, while *S. cerevisiae* is unable to consume lactose and could be as secondary contaminant which uses sugars released by other microbes, *T. delbrueckii* and *W. pararugosa* strains exhibit a β-galactosidase activity to assimilate lactose. The genetic basis of this specialized trait required further investigations in the future to understand how these strains adapted to dairy environments.

The pool of fast-fermenting *K. marxianus* strains identified in this study can valorize the sugar fraction of whey as they reached ethanol yield from 83.8% to app. 100%. Remarkably, strain CA116 exhibits both thermotolerance until 48 °C and high ethanol production, emerging as the best candidate for ethanol production from whey. Whey contains proteins other than sugar and can be utilized as raw material for producing beverages and hydrolysates enriched in bioactive peptides. This option has been mainly investigated through LAB fermentation, with a few attempts using yeasts. Here we demonstrated that *K. marxianus* and *T. delbrueckii* can release whey-derived bioactive peptides mainly with anti-hypertensive activity, opening a new avenue for the utilization of these strains as bioactive-peptide producers. Interestingly *K. marxianus* strain CA214 exhibits high ethanol yield and strongly ability to hydrolyze proteins releasing the highest amount of bioactive and ACE-inhibitory peptides. As reported above, identification of proteases responsible for this phenotype could assist future selection program and shed light on why these yeasts require proteolytic activity to survive in dairy environment.

## Figures and Tables

**Figure 1 microorganisms-09-02288-f001:**
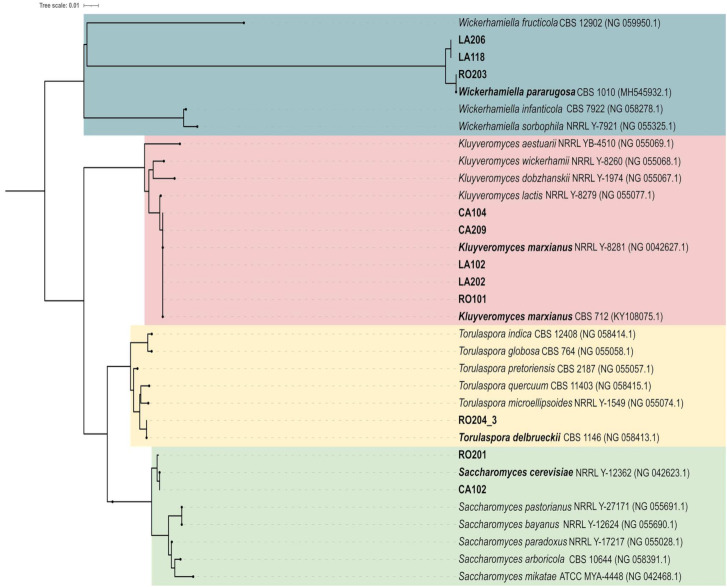
The neighbor joining (NJ) tree inferred from the dataset containing the 33 26S rRNA D1/D2 nucleotide sequences. The GenBank accession numbers were reported in the brackets. PR NWS yeast strains are reported in bold. The evolutionary distances were computed using the Kimura 2-parameter method and are in the units of the number of base substitutions per site. The rate variation among sites was modeled with a gamma distribution. Branch lengths are proportional to the numbers of nucleotide substitutions and are measured by the scale bar of sequence divergence.

**Figure 2 microorganisms-09-02288-f002:**
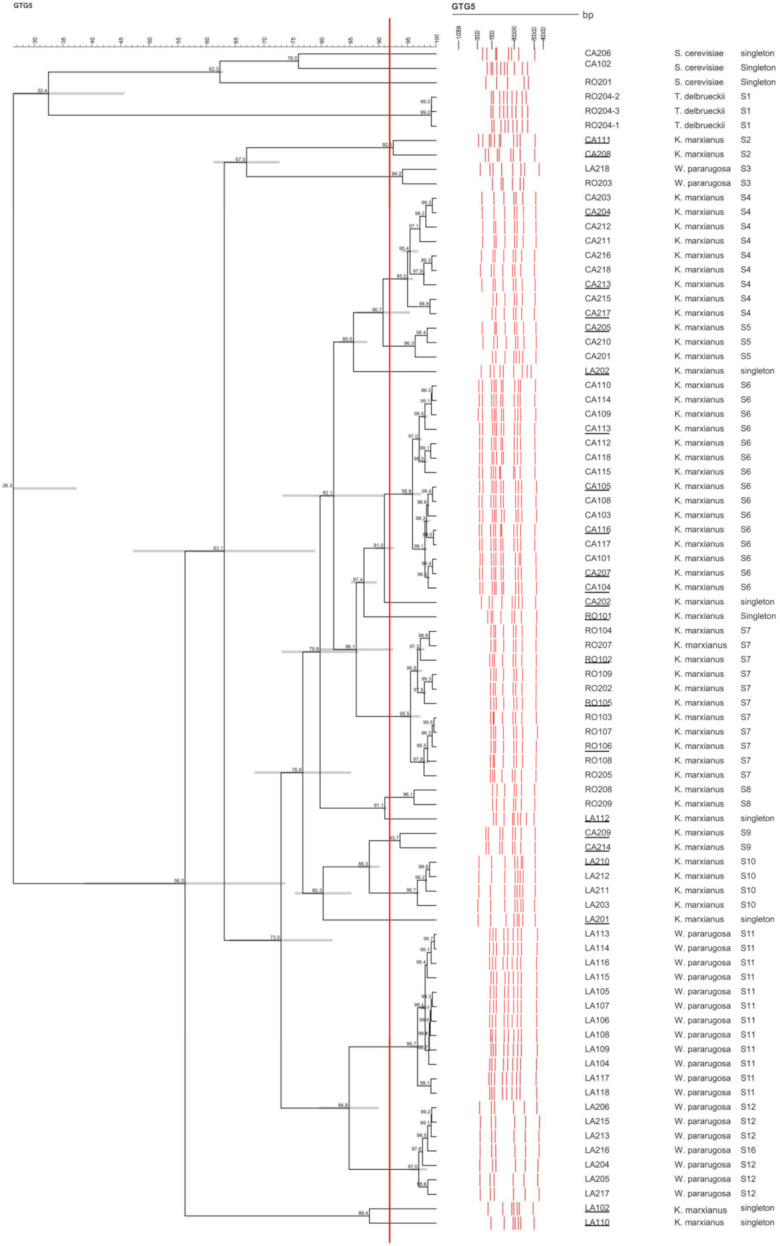
Tree obtained from UPGMA analysis of (GTG)_5_ MSP-PCR profiles of 82 NWS yeast isolates, using Pearson’s correlation coefficient. The similarity value of 92% was used for biotype discrimination. Selected strains are reported in bold. Error bars are reported in grey at each node together with the percentage of similarity.

**Figure 3 microorganisms-09-02288-f003:**
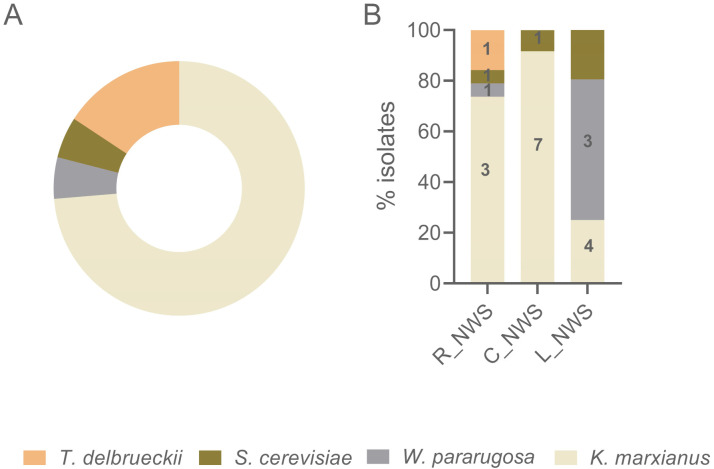
Yeast species frequencies and distribution in PR NSW samples. Pie-chart (panel (**A**)) represents yeast species frequencies, while column graph (panel (**B**)) species distribution in each sample. Numbers on the column represent biotypes scored by UPGMA analysis of (GTG)_5_ MSP-PCR fingerprinting data.

**Figure 4 microorganisms-09-02288-f004:**
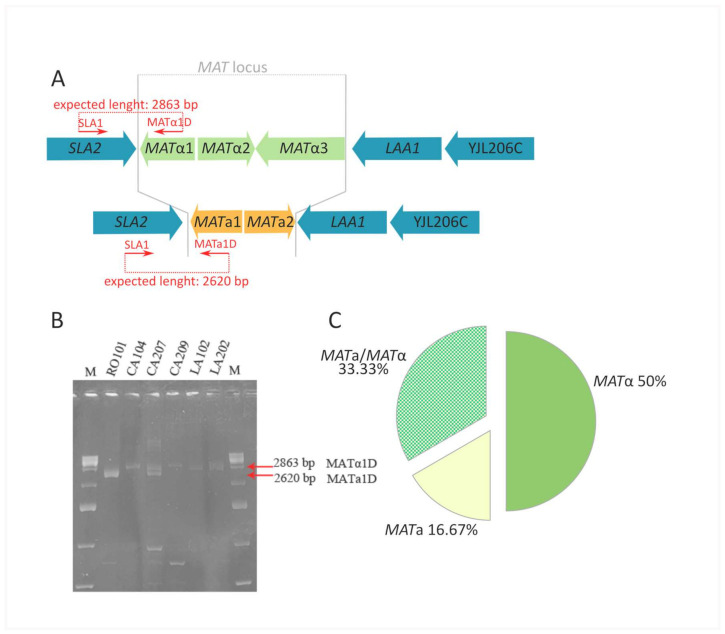
*MAT* genotyping of *Kluyveromyces marxianus* strains. (**A**) Primer sets anneal the either Ya or Yα region in the *MAT* locus and the flanking gene *SLA2* outside the Z region. PCR amplicons of different length discriminates *MAT*a and *MAT*α loci (panel (**A**)). (**B**) An example of PCR products obtained using multiplex PCR assay targeting *SLA2*-*MAT* junctions. (**C**) Percentage of *MAT* genotypes is summarized in pie chart.

**Figure 5 microorganisms-09-02288-f005:**
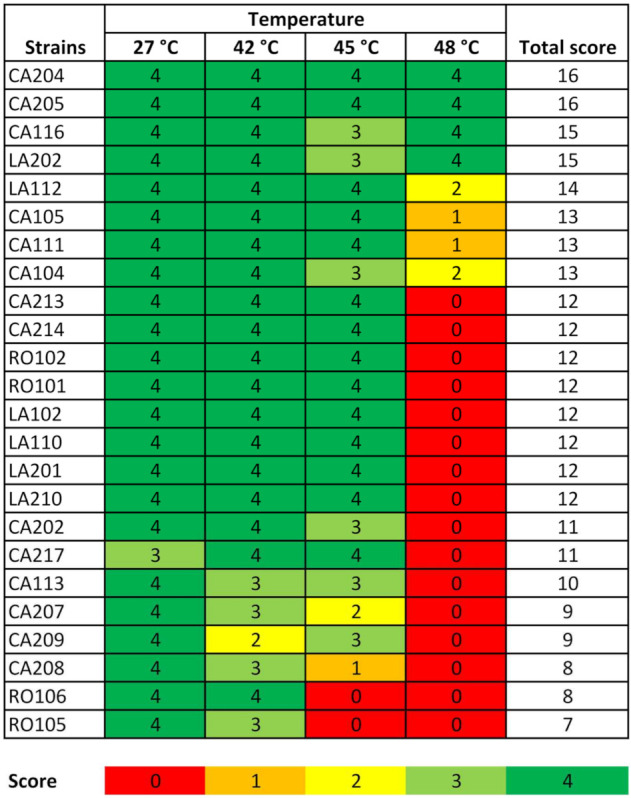
Growth of *K. marxianus* strains at different temperatures (27, 42, 45, and 48 °C). Scores were obtained as the total number of spots for which growth at each dilution was evident. A heatmap-like color scale was adopted as a proxy for numerical values varying from red (no growth, or 0 points) to dark green (four spots/dilution). Strains were ordered based on decreasing total score.

**Figure 6 microorganisms-09-02288-f006:**
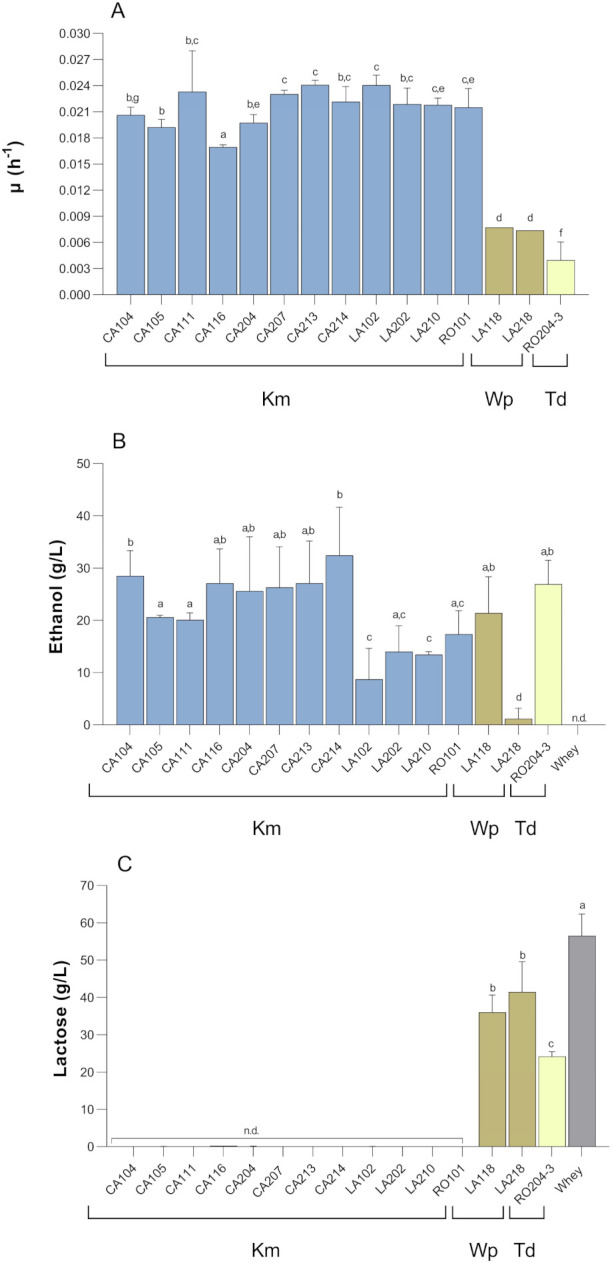
Growth rate (**A**), ethanol production (**B**), and lactose utilization (**C**) in whey fermentation trials. Details on strain isolation are in Table 2. Different letters mean significant differences (*p* < 0.05) among strains. Data are represented by the mean (*n* = 3); error bars show standard deviation. Abbreviations: Km, *K. marxianus*; Wp, *W. pararugosa*; Td, *T. delbrueckii*; nd, not detected.

**Figure 7 microorganisms-09-02288-f007:**
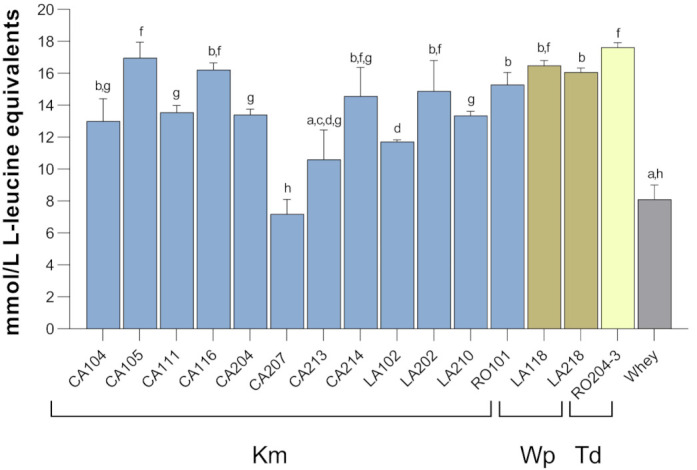
Proteolytic activity in fermented whey. The amount of released amino group was expressed as mmol/L of leucine equivalents. Details on strain isolation are in Table 2. Different letters mean significant differences (*p* < 0.05) among strains. Data are represented as mean (*n* = 3); error bars show standard deviation. Abbreviations: Km, *K. marxianus*; Wp, *W. pararugosa*; Td, *T. delbrueckii*; nd, not detected.

**Figure 8 microorganisms-09-02288-f008:**
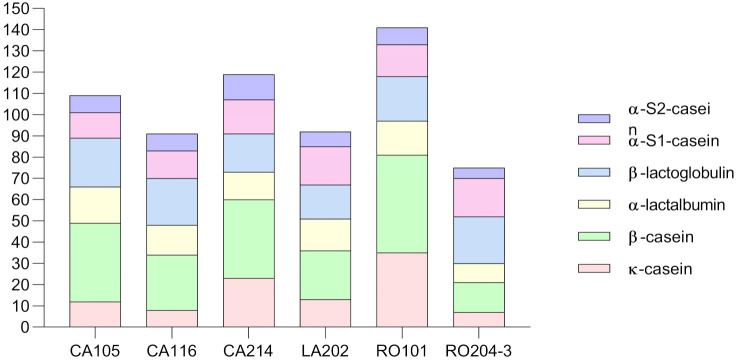
Number of identified peptides by mass spectrometry in fermented whey trials. Peptides are reported as function of the protein type. Details on strain isolation are in Table 2.

**Figure 9 microorganisms-09-02288-f009:**
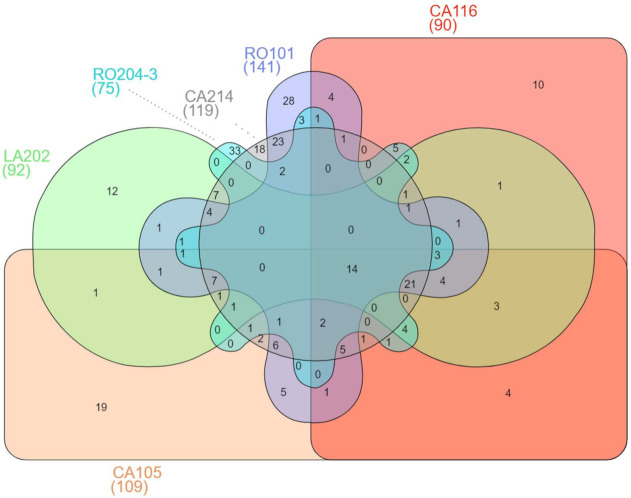
Venn diagram showing differences between the pattern of peptides released after whey fermentation by selected yeast strains. The complete list of peptides identified at the end of the fermentation trials by mass spectrometry can be found in Appendix A. Details on strain isolation are in Table 2.

**Table 1 microorganisms-09-02288-t001:** Physicochemical and microbiological parameters of NWS samples. Values (Log10 CFU/mL) are means ± SD of three replicates (n = 3).

Sample	pH	SH	LAB Count (Log_10_ CFU/mL)	Yeast Counts (Log_10_ CFU/mL)	
				YPDA 28 °C	YPLA 42 °C
R_NWS	3.29 ± 0.01 ^a^	31.01 ± 0.02 ^a^	8.65 ± 0.01 ^a^	2.45 ± 0.11 ^a,^*	2.03 ± 0.06 ^a,^*
C_NWS	3.27 ±0.00 ^a^	31.41 ± 0.01 ^c^	8.71 ± 0.01 ^b^	2.74 ± 0.04 ^a^	2.84 ± 0.06 ^a^
L_NWS	3.44 ±0.00 ^b^	31.37 ± 0.01 ^b^	8.72 ± 0.01 ^b^	3.43 ± 0.32 ^b^	3.41 ± 0.21 ^b^

Values within a column with different superscript letters are significantly different (*p* < 0.05). Asterisks, when present, indicate significant differences between growth conditions (between columns) (*p* < 0.05).

**Table 2 microorganisms-09-02288-t002:** Isolation source and ITS restriction patterns of PR NWS yeast isolates. Tentative yeast identification was obtaining by using ITS restriction patterns as queries in Yeast ID databases (www.yeast-id.org; last accessed on 9 September 2021).

Sample	Growth Conditions	Strains	Amp (bp)	Restriction Fragments		Yeast ID Best Matches	Pattern
				*Hae*III	*Hinf*I		
**R_NWS**	**YPDA, 28 °C**	**RO101 RO102 RO103 RO104 RO105 RO106 RO107 RO108 RO109**	**740**	**655-80**	240-185-120-80	*K. dobzhanskii* (100%)/ *K. marxianus* (94%)	A
	YPLA 42 °C	RO202 RO205 RO207 RO208 RO209	740	655-80	240-185-120-80	*K. dobzhanskii* (100%)/ *K. marxianus* (94%)	A
		**RO201**	850	325-230-170-125	375-365-110	*S. cerevisiae* (100%)	B
		**RO203**	420	420	200-220	*Candida spp.* (100%)	C
		RO204-1 RO204-2 **RO204-3**	800	800	425-400	*T. delbrueckii* (100%)	D
C_NWS	YPDA, 28 °C	CA101 CA103 **CA104** CA105 CA107 CA108 CA109 CA110 CA111 CA112 CA113 CA114 CA115 CA116 CA117 CA118	740	655-80	240-185-120-80	*Kluyveromyces dobzhanskii* (100%)/*Kluyveromyces marxianus* (94%)	A
		**CA102** CA106	850	325-230-170-125	375-365-110	*S. cerevisiae* (100%)	B
	YPLA 42 °C	CA201 CA202 CA203 CA204 CA205 CA207 CA208 **CA209** CA210 CA211 CA212 CA213 CA214 CA215 CA216 CA217 CA218	740	655-80	240-185-120-80	*K. dobzhanskii* (100%)/ *K. marxianus* (94%)	A
		CA206	850	325-230-170-125	375-365-110	*S. cerevisiae* (100%)	B
L_NWS	YPDA, 28 °C	**LA102** LA110 LA112	740	655-80	240-185-120-80	*K. dobzhanskii* (100%)/ *K. marxianus* (94%)	A
		LA101 LA103 LA111	850	325-230-170-125	375-365-110	*S. cerevisiae* (100%)	B
		LA104 LA105 LA106 LA107 LA108 LA109 LA113 LA114 LA115 LA116 LA117 **LA118**	420	420	200-220	*Candida spp.* (100%)	C
	YPLA 42 °C	LA201 **LA202** LA203 LA210 LA211 LA212	740	655-80	240-185-120-80	*K. dobzhanskii* (100%)/ *K. marxianus* (94%)	A
		LA207 LA208 LA209 LA214	850	325-230-170-125	375-365-110	*S. cerevisiae* (100%)	B
		LA204 LA205 **LA206** LA213 LA215 LA216 LA217 LA218	420	420	200-220	*Candida spp.* (100%)	C

Strains in bold were submitted to sequencing of 26S rDNA D1/D2 domain, while underlined strains were screened for whey fermentation. All the fragment sizes are in bp. Fragments lower than 70 bp were omitted from the analysis. Abbreviation: Amp, amplicon length.

**Table 3 microorganisms-09-02288-t003:** Peptides scored in fermented whey with 100% sequence homology with previously demonstrated bioactive peptides. Abbreviation: ACE, angiotensin I-converting enzyme.

Protein Precursor	Peptide Sequence	Biological Activity	Strains
β-casein	VYPFPGPIPN	Antioxidant, ACE-inhibitory	CA105, CA116, LA202, RO101
	SLPQ	ACE-inhibitory	CA214
	VVPP	ACE-inhibitory	CA214
	EAMAPK	Antimicrobial	CA214
	LHLPLP	ACE-inhibitory	CA105, CA116, CA214, LA202, RO101
	HQPHQPLPPT	ACE-inhibitory	CA105, CA116
	SQSKVLPVPQ	ACE-inhibitory	CA105, CA116, CA214, LA202, RO101, RO204-3
	SQSKVLPVPQKAVPYPQ	Antioxidant	RO204-3
	SKVLPVPQ	ACE-inhibitory	CA214
	KVLPVP	ACE-inhibitory	CA214, RO101
	KVLPVPQ	ACE-inhibitory, Anti-inflammatory	CA105, CA214, RO101
	KVLPVPQK	Antioxidant	RO101
	YQEPVLGPVR	Antioxidant, ACE-inhibitory, Anti-inflammatory, Antithrombotic, Immunomodulatory	CA105, CA214, LA202, RO101
α-S1-casein	RPKHPIKHQ	ACE-inhibitory	CA105, CA214, LA202, RO204-3
	LRLKKYKVPQL	Antimicrobial	CA116, LA202, RO204-3
	PEL	Antioxidant	CA214, LA202
	SDIPNPIGSENSEK	Antimicrobial	RO204-3
κ-casein	VESTVATL	Antimicrobial	CA214, RO101
α-lactalbumin	YGL	ACE-inhibitory	CA105, RO101
	DKVGINYW	ACE-inhibitory	CA214, RO101
β-lactoglobulin	AVF	Anti-inflammatory	CA116, RO204-3
	VLVLDTDYK	DPP-IV Inhibitory, Antimicrobial	CA214**,** RO101

## Data Availability

The nucleotide sequences of 26S rDNA D1/D2 domain are available in GenBank NCBI database under the accession numbers MZ491084 to MZ491094.

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
