# Peer review of "Characterization of Yeasts Isolated from Parmigiano Reggiano Cheese Natural Whey Starter: From Spoilage Agents to Potential Cell Factories for Whey Valorization"

_microorganisms, 2021, doi:10.3390/microorganisms9112288_

Round 1
Reviewer 1 Report
The topic and the approach of the study are very interesting and a great deal of work was conducted. A large part seems to be dedicated to the evaluation of molecular analysis and bioinformatics, and maybe should be noted in the abstract. Regarding the fermentation part, though, I am afraid that there are some ambiguous points that have to be checked and clarified. Some specific comments are provided below
Lines 19-20 “consists of the well-documented Kluyveromyces marxianus, as well as of other species (Saccharomyces cerevisiae, Wickerhamiella pararugosa, and Torulaspora delbrueckii) …
Line 24: what does the yield stand for. Does it refer to fermentation efficiency (% of the maximum theoretical)?. Please elucidate
Line 25: Another candidate? Please provide the strain
Line 47: Please rephrase
Line 50: please rephrase
Line 53: bio-products instead of bio-commodities
Line 83: please provide reference
Line 101: 50 ml
Line 203: 5 ml
Lines 215-216: How accurate can this be especially since the culture volume was only 50 ml? Moreover at what degree does CO2 pass through paraffin oil?
Line 217: Isn’t 14 days too long for yeast fermentation?
Lines 249 and 252: What is meant by the cultivable fraction of yeasts? Did you also measure the whole yeast population including the non-cultivable strains? If not please remove the term fraction
Lines 386 and 388: It is quite surprising that such high titters of ethanol were measured since according to the concentration of sugars in the whey (lines 211—212) (and bearing in mind that during alcoholic fermentation the maximum ethanol yield from hexoses (glucose, galactose) is 0.51g/g and for disaccharides (lactose) 0.53 g/g) a theoretical maximum of 23g/L could only be achieved, i.e practically up to 20g/L since sugars are used for the production of biomass and possibly other metabolites.
Line 393-397: I believe that it is 0.107 mol/L (36.7g/L / 342 mol/g) and not 0.17, according to which complete fermentation of lactose should result to 0.5 mol/L ethanol. Please check your calculations and analysis of fermentation broths again.
Figure 6: As in the previous comment please check the accuracy of the results presented for ethanol production, In all cases the titters are very high. For LA118 and RO204-3 where did the ethanol come from if lactose was not consumed at all? Moreover why the concentration of lactose in whey (control) is 55g/L and in the LA218 culture above 40g/L. The lactose content in the M& M section is given as 3.67% w/v.
Author Response
RESPONSE TO CONCERNS ARISEN BY REVIEWER 1
Review 1. The topic and the approach of the study are very interesting and a great deal of work was conducted. A large part seems to be dedicated to the evaluation of molecular analysis and bioinformatics, and maybe should be noted in the abstract.
Reply. Abstract has been modified to better describe molecular approaches.
Review 1. Lines 19-20 “consists of the well-documented Kluyveromyces marxianus, as well as of other species (Saccharomyces cerevisiae, Wickerhamiella pararugosa, and Torulaspora delbrueckii) …
Reply. Done
Review 1. Line 24: what does the yield stand for. Does it refer to fermentation efficiency (% of the maximum theoretical)?. Please elucidate
Reply. We apologize for the mistake. We intended fermentation efficiency, ie. measure of how much alcohol is finally produced relative to the amount that could be theoretically produced.
Review 1. Line 25: Another candidate? Please provide the strain
Reply. Done.
Review 1. Line 47: Please rephrase
Reply. Done (Lines 47-50, page 2 in the revised manuscript).
Review 1. Line 50: Please rephrase
Reply. Done (Lines 50-53, page 2 in the revised manuscript).
Review 1. Line 53: bio-products instead of bio-commodities.
Reply. Done (Line 56, page 2 in the revised manuscript).
Review 1. Line 83: please provide reference.
Reply. Done (Line 85, page 2 in the revised manuscript).
Review 1. Line 101: 50 ml
Reply. Done (Line 104, page 3 in the revised manuscript).
Reviewer 1. Line 203: 5 ml
Reply. Done (Line 204, page 5 in the revised manuscript).
Reviewer 1. Lines 215-216: How accurate can this be especially since the culture volume was only 50 ml? Moreover at what degree does CO2 pass through paraffin oil?
Reply. We usually performed CO2 weight loss assays as conventional method to measure fermentation kinetics using CO2 released as weight loss. Generally, we performed weight loss fermentation tests in 100 mL of medium, but in this specific case we scaled down the volume to 50 mL to have enough acidic whey to perform all the tests under the same experimental conditions. Our preliminary assay showed that no significant differences were found by reducing fermentation volume from 100 to 50 mL. Regarding paraffin oil, this is usually exploited in weight loss fermentation tests as reported in Tofalo et al. (2014) and in many other papers (Romano et al. 2003; Cioch-Skoneczny et al. 2020; Speranza et al. 2010), to avoid that air exposition can affect the CO2 weight loss due to fermentative pathway.
Reviewer 1. Lines 249 and 252: What is meant by the cultivable fraction of yeasts? Did you also measure the whole yeast population including the non-cultivable strains? If not please remove the term fraction.
Reply. We remove this term.
Reviewer 1. Line 217: Isn’t 14 days too long for yeast fermentation?
Reply. Timing of fermentation agreed with that performed by Tofalo et al (2014). In the set of experiments shown in this paper, we considered end-point measurements of ethanol and protein hydrolysis. We stopped fermentations assays when weight loss values did not change for at least two consecutive measurements, or arbitrarily, after 14 days of fermentation in the case of slow-fermenting strains. Further experiments are currently ongoing to evaluate proteolysis degree and ethanol production over time and to find a balance between proteolysis and timing of fermentation.
Reviewer 1. Lines 249 and 252: What is meant by the cultivable fraction of yeasts? Did you also measure the whole yeast population including the non-cultivable strains? If not please remove the term fraction
Reply. We removed the term.
Reviewer 1. Lines 386 and 388: It is quite surprising that such high titters of ethanol were measured since according to the concentration of sugars in the whey (lines 211—212) (and bearing in mind that during alcoholic fermentation the maximum ethanol yield from hexoses (glucose, galactose) is 0.51g/g and for disaccharides (lactose) 0.53 g/g) a theoretical maximum of 23g/L could only be achieved, i.e practically up to 20g/L since sugars are used for the production of biomass and possibly other metabolites.
Reply. We apologize for the mistaken calculation. Acidic whey used for fermentation trials contained 56.7 g/L lactose and not 36.7 g/l as erroneously reported in material and methods. We probably did a typo error during the writing of the manuscript. Indeed, the hexoses (glucose + galactose) amount in whey was 7.0 g/L. Taking into account the maximum ethanol yield during alcoholic fermentation (0.53 g/g for lactose and 0.51 g/g for hexoses), a theoretical maximum ethanol amount of 33.7 g/l could be achieved. Strain CA241 produced 32.5 g/l of ethanol, which corresponded to a 96.3% of fermentation efficiency. According to the reviewer’s suggestion, we modified this part of the manuscript (please see lines 390-399 page 12 of the revised manuscript).
Reviewer 1. Line 393-397: I believe that it is 0.107 mol/L (36.7g/L / 342 mol/g) and not 0.17, according to which complete fermentation of lactose should result to 0.5 mol/L ethanol. Please check your calculations and analysis of fermentation broths again.
Reply. The calculations were right. The amount of lactose in whey was 56.7 g/L and not 36.7 g/l as erroneously reported in material and methods, which resulted in a molar concentration of 0.17 mol/L. Anyway, this part of the manuscript has been revised according to the previous reviewer’s comment (please see lines 390-399 page 12 of the revised manuscript).
Reviewer 1. Figure 6: As in the previous comment please check the accuracy of the results presented for ethanol production, In all cases the titters are very high. For LA118 and RO204-3 where did the ethanol come from if lactose was not consumed at all? Moreover why the concentration of lactose in whey (control) is 55g/L and in the LA218 culture above 40g/L. The lactose content in the M& M section is given as 3.67% w/v.
Reply. Once again, as now reported in material and methods and coherently with Figure 6, the lactose whey concentration was 56.7 g/L and not 36.7 g/L. We are very sorry for this typo error which created a misunderstanding in ethanol calculation. Considering the total lactose and hexoses amount in whey, the ethanol efficiency for most of the strains was below the maximum amount achievable considering a conversion factor of 0.53 g/g for lactose and 0.51 g/g for hexoses. Concerning W. pararugosa LA118 and T. delbrueckii RO204-3, we also reported in the manuscript that these strains produced ethanol amounts which exceed those expected from the lactose consumption. We also hypothesized that amino acids released from whey protein hydrolysis could be used both for TCA cycle and pyruvate production (followed by reduction to ethanol) (lines 426-430 in the revised manuscript). Another hypothesis could be that, in absence of any other fermentable sugars, these yeasts can convert lactic acid into ethanol, similarly to S. cerevisiae (Wakamatsu et al. 2013). However, these metabolic pathways were poorly investigated in T. delbrueckii and W. pararugosa, so we preferred to omit this last hypothesis in the main text. The reason why these strains produced so many titers of ethanol is currently under study in our lab.
References
Cioch-Skoneczny M. et al. (2020) European Food Research and Technology 246, 2299–2307.
Tofalo R. et al. (2014) International Journal of Food Microbiology 187, 41–49.
Romano et al. (2003) International Journal of Food Microbiology 86, 169–180.
Speranza et al. (2019) Fermentation 5(4), 102; https://doi.org/10.3390/fermentation5040102.
Wakamatsu M et al. (2013) Journal of Bioscience and Bioengineering 116, 85-90.

Reviewer 2 Report
I find this manuscript to be very well written with a well designed experimental approach. I have only one minor comment for Table 1: I think you forgot one rather important detail in the table, since I would assume all values for counts/CFU are log10?
Author Response
RESPONSE TO CONCERNS ARISEN BY REVIEWER 2
Review 2. I find this manuscript to be very well written with a well designed experimental approach. I have only one minor comment for Table 1: I think you forgot one rather important detail in the table, since I would assume all values for counts/CFU are log10?
Reply. Yes, values were log 10 CFU/mL. We added this information to table and Table caption (line 255, page 6 in the revised manuscript).

Round 2
Reviewer 1 Report
The revision of the MS and the clarifications provided by the authors are efficient